# Characterization of the *Aspergillus flavus* Population from Highly Aflatoxin-Contaminated Corn in the United States

**DOI:** 10.3390/toxins14110755

**Published:** 2022-11-02

**Authors:** Mark A. Weaver, Kenneth A. Callicott, Hillary L. Mehl, Joseph Opoku, Lilly C. Park, Keiana S. Fields, Jennifer R. Mandel

**Affiliations:** 1USDA-ARS National Biological Control Laboratory, Stoneville, MS 38776, USA; 2USDA-ARS, Tucson, AZ 85701, USA; 3USDA-ARS National Center for Agricultural Utilization Research, Peoria, IL 61604, USA; 4Department of Biological Sciences, University of Memphis, Memphis, TN 38152, USA

**Keywords:** population genetics, mycotoxin, mating type, fungicide resistance, maize

## Abstract

Aflatoxin contamination of corn is a major threat to the safe food and feed. The United States Federal Grain Inspection Service (FGIS) monitors commercial grain shipments for the presence of aflatoxin. A total of 146 *Aspergillus flavus* were isolated from 29 highly contaminated grain samples to characterize the visual phenotypes, aflatoxin-producing potential, and genotypes to explore the etiological cause of high aflatoxin contamination of US corn. Five of the isolates had reduced sensitivity (43–49% resistant) to the fungicide azoxystrobin, with the remainder all being over 50% resistant to azoxystrobin at the discriminating dose of 2.5 µg/mL. Only six isolates of the highly aflatoxigenic *S* morphotype were found, and 48 isolates were non-aflatoxigenic. Analysis of the mating type locus revealed 45% MAT 1-1 and 55% MAT 1-2. The *A. flavus* population originating from the highly aflatoxin contaminated grain samples was compared to a randomly selected subset of isolates originating from commercial corn samples with typical levels of aflatoxin contamination (average < 50 ppb). Use of simple sequence repeat (SSR) genotyping followed by principal component analysis (PCoA) revealed a similar pattern of genotypic distribution in the two populations, but greater diversity in the FGIS-derived population. The noticeable difference between the two populations was that genotypes identical to strain NRRL 21882, the active component of the aflatoxin biocontrol product Afla-Guard™, were ten times more common in the commercial corn population of *A. flavus* compared to the population from the high-aflatoxin corn samples. The other similarities between the two populations suggest that high aflatoxin concentrations in corn grain are generally the result of infection with common *A. flavus* genotypes.

## 1. Introduction

*Aspergillus flavus* is a common soil-borne saprophyte and opportunistic pathogen of animals and plants. Crops attacked by *A. flavus* do not suffer significant yield loss but may become contaminated with aflatoxin B_1_ [1], the most potent carcinogenic compound occurring in nature [2]. Yearly aflatoxin-related liver cancer deaths have been estimated to be over 100,000, especially in low- and middle-income countries where corn and peanuts are dietary staples [3]. Aflatoxins (Afs), including aflatoxin B_1_, B_2_, G_1_, and G_2_, are monitored in susceptible crops including corn, peanuts, and tree nuts, and their presence is regulated to various extents by governments around the world. AF contamination in corn above 20 ppb is subject to regulation by the US Department of Agriculture and Food and Drug Administration, restricting its use and, thus, excluding highly contaminated grain from the market [4]. Grain shipments are tested repeatedly at different points as commodities move through the supply chain, ensuring the safety of food and feed [5]. In the United States, the Federal Grain Inspection Service (FGIS) provides inspection services and information on the proper use of sampling and inspection equipment, application of the grain standards, and availability of official inspection services for grains, pulses, oilseeds, and processed and graded commodities. These services facilitate the efficient and effective marketing of US grain and other commodities from farmers to domestic and international end users. 

The AF surveillance programs are highly effective but come at a substantial cost. In addition to the burden of testing itself, the economic loss due to AF in 16 states from 2001 to 2016 averaged 17.5–24.5 million USD [6]. Biotic and abiotic factors that influence aflatoxin contamination may include high temperatures, high humidity, drought, insect damage, host susceptibility, and the presence of highly aflatoxigenic genotypes of *A. flavus* [7,8,9]. Agronomic practices, hybrid selection, transgenic insect protection, and irrigation can provide a degree of protection against AF contamination [10,11,12], but the most reliably effective and economically viable protection is the field application of non-aflatoxigenic (atoxigenic) biocontrol strains of *A. flavus* [3,13], such as Afla-Guard^®^ and AF36 Prevail^®^, commercial formulations of strains NRRL 21882 and NRRL 18543, respectively. The atoxigenic *A. flavus* biocontrol approach has been used in maize, groundnuts, and cottonseed worldwide [14,15]. AF contamination of corn is highly variable, and biocontrol application must be made early in the growing season before the risk for a particular season and field is well known. One risk factor that might be assessed early in the growing season is the aflatoxigenic potential of the *A. flavus* population in a particular field. Aflatoxin-producing potential is variable within *A. flavus* populations, with some genotypes producing high concentrations of AF and others producing none [9,16]. Two morphological types described within *A. flavus* (L and S strain) exemplify this variation, with the S strain producing consistently large quantities of AF, whereas some L strains have lost the capacity to produce AF entirely [17]. Numerous studies have established that even relatively low frequencies of highly aflatoxigenic genotypes can contribute to high aflatoxin concentrations in the crop. There is great diversity within the *A. flavus* population in aflatoxigenic potential, and some genotypes appear to be more commonly associated with infected corn [16]. The aflatoxin-producing potential of field populations of *A. flavus* that become associated with the crop contributes to the risk of high aflatoxin concentrations in the harvested grain. Thus, *A. flavus* genotypes infecting highly contaminated grain may have greater aflatoxin-producing potential compared to *A. flavus* populations infecting corn with low to moderate aflatoxin concentrations.

In the present survey, we obtained corn samples from FGIS with unusually high (>50 ppb) AF contamination. Some visual phenotypes have been associated with high virulence or toxigenicity [17,18]; thus, isolates from the highly contaminated corn samples (FGIS isolates) were scored for those phenotypes to see if the same relationships were found in this unique set of samples. Another phenotype of concern is fungicide resistance; emergent fungicide resistance within a population of *A. flavus* in the southeastern US was recently documented and is a concern for peanut production [19], thus prompting an interest in how common this resistance is within other *A. flavus* populations. Our objective was to document the visual phenotypes, fungicide tolerance, and genotypic diversity of *A. flavus* isolates associated with the most highly AF-contaminated corn in the US. In addition to phenotyping isolates associated with the highly contaminated corn samples, genotypes within this population of isolates were compared to isolates from a random sampling of commercial corn with more typical levels of aflatoxin contamination to test the hypothesis that the isolates from high AF corn are phenotypically and/or genetically distinct. 

## 2. Results and Discussion

### 2.1. Phenotypes of Isolates from High-AF Corn

*Aspergillus flavus* was isolated from 29 samples of highly AF-contaminated (>50 ppb) corn from FGIS. The visual phenotypes of the 146 *A. flavus* strains isolated from the highly AF-contaminated corn are presented in Figure 1. About one-third of the FGIS isolates (*N* = 146) fluoresced under UV light when grown on a medium with a fluorescence enhancer (Figure 1A), and 61% of the isolates produced a dark-yellow-brown pigment on the colony reverse (Figure 1B), traits associated previously with aflatoxigenicity [18]. While previous reports on these cultural markers showed a linkage between the yellow phenotype and fluorescence and that together they were a strong predictor of aflatoxigenicity, the linkage between these phenotypes and aflatoxigenicity was less robust in the present study. Here, we noted about a 20% error rate (combined false positives and false negatives) for predictions of aflatoxigenicity by fluorescence and over 30% error rate for predictions based on yellow pigment. An important difference is that, while the earlier study included a larger number of isolates, they were from a very limited geographic area. That area, the Mississippi Delta, has since been shown to be well colonized by NRRL 21882, a “white, nonfluorescent” type, or 21882-like genotypes.

S-type morphology was not common in the present collection (Figure 1C), accounting for only six of the isolates. The S-type isolates, however, were all aflatoxigenic, validating their role as a reliable predictor of aflatoxigenicity. The low abundance of S-type isolates is consistent with the collection of corn-derived isolates in Louisiana [16] but differs from the collections on Texas corncobs (11–66%) [20] and on some soil samples [20,21]. Reports from Africa of corn with exceptionally high aflatoxin contamination, including several over 1 µg/mL, were linked to especially virulent S-type strains, with the level of contamination increasing with an increased proportion of S-type infection [22]. However, the S-type strains at issue in the African dataset [22] were not *A. flavus* but were from a related but distinct clade of aflatoxin B and aflatoxin B/G producers. S-type isolates in the US are almost entirely *A. flavus* [23], as was the case in this study.

Following observations of poor peanut seed germination in Georgia, USA, it was discovered that *A. flavus* from these seed lots had mutations in the cytochrome b gene conferring resistance to the quinone outside inhibitor (QoI) fungicide azoxystrobin [19]. Azoxystrobin is not labeled for management of *A. flavus* infection of corn, but the fungicide is used in commercial corn production and is labeled for *Aspergillus* diseases in other crops. The intensive use in many crops means that the fungicide is somewhat commonplace in the US agroecosystem, and *A. flavus* has had substantial exposure to the fungicide. In fact, our observations of *A. flavus* (Figure 2) from these highly AF-positive corn samples indicate a high degree of tolerance to azoxystrobin at a discriminatory dose that was highly inhibitory to the susceptible *A. flavus* in the Georgia peanut population. The resistance found in these FGIS isolates differed from the Georgia peanut isolates in that 50% of the Georgia isolates were “sensitive” or had “reduced sensitivity”, while none of the FGIS isolates were “sensitive”, 3% had “reduced sensitivity”, and 97% were “moderately resistant” or “resistant” according to the criteria of Ali et al. [19]. This may suggest regional or crop-associated differences in selection pressure for resistance to QoI fungicides in *A. flavus* populations.

### 2.2. Assignment of Genotype and Chemotype to Isolates from High-AF Corn and Comparison to Isolates from Commercial Corn

Evidence for niche specialization in *A. flavus* populations has been presented previously [16]. An important aspect of that specialization is that, while soil populations in the previous study were evenly split between MAT 1-1 and MAT 1-2, the population on infected corn was 96% MAT 1-2. The isolates in the present study were much more evenly divided with 55% MAT 1-2 (Figure 3). The present results do not directly contradict the earlier findings due to important differences in the source materials. The results of the earlier work may have reflected the population of that limited geographic area and limited sampling period, in contrast with the much larger area and longer sampling time of the present study. More importantly, the isolates examined here come from a unique set of corn samples, with AF contamination higher than 99% of US corn samples. Thus, it is entirely possible that the mating type found on US corn samples, generally, is largely composed of MAT 1-2, but AF contamination is associated with an enrichment of MAT 1-1 genotypes.

In addition to the visual phenotype, the chemotype of each isolate was determined (Figure 4) after growing on moistened, autoclaved corn. While the aflatoxin-producing potential of *A. flavus* isolates has been measured on numerous synthetic or semisynthetic media, production on autoclaved corn has been shown to be the most reliable [24]. Even in these optimized conditions, about half of the isolates in this study produced low (<20 ng/mL) or zero AF (LOQ 1 ng/mL). It seems unlikely that these isolates would lead to corn contaminated with AF above the 20 ng/g regulatory threshold under field conditions. Instead, the causative agents of the high-AF corn studied here are most likely isolates identified here as medium- or high-producing, and previous studies have reported that, even when highly aflatoxigenic genotypes are present at relatively low frequencies within the population, their presence can lead to high overall aflatoxin concentrations in the crop [25]. The isolates in this study were from grain samples that were unusually highly AF-contaminated, and it might be expected that these isolates differed from a more typical sampling of commercial corn isolates. In fact, less than one-fourth of the corn isolates from Sweany et al. [16] were medium- or high-producing isolates, while in the present study 40% were medium- or high-producing isolates.

Examination of the alleles at 15 SSR (simple sequence repeat) loci revealed two broad similarities and one important difference between the FGIS isolates in the present study and the isolates from a more general US survey of commercial corn samples. First, when analyzed with principal coordinate analysis (PCoA), the two populations generally followed a similar distribution pattern (Figure 5). Furthermore, aflatoxigenic and non-aflatoxigenic individuals from both the FGIS and the commercial corn populations showed high genetic similarity to strain NRRL 21882, the active ingredient of the commercial biocontrol product Afla-Guard^®^. Interestingly, half (75 of 150) of the commercial corn isolate population had genotypes that were identical to NRRL 21882, while only seven of the 146 FGIS isolates were that genotype. Recognizing the limitations of these two modest-sized samples, this is compelling and unique evidence for the efficacy of isolate NRRL 21882 in preventing high-AF contamination of corn. The high frequency of 21882-like, but not identical, isolates may also address the evolutionary origins of NRRL 21882 [26]. It is possible that these common 21882-like isolates are derived from NRRL 21882, which has been applied in many US corn fields. This would be highly concerning, as several of these 21882-like isolates are aflatoxigenic (data not shown). More likely is that this group is a very common, successful *A. flavus* clade that has retained similar, albeit not identical, SSR patterns even as the AF trait has been variously retained or lost in different lineages. Using multilocus sequence haplotyping, Lewis et al. [27] found a lineage, which included 21882-like haplotypes, that was dominant in agricultural soils of three southeastern US states before and after application of biocontrol strains. Previous studies have shown that non-aflatoxigenic isolates are more closely related to aflatoxigenic isolates than they are to other non-aflatoxigenic groups [28,29,30].

To compare across FGIS samples and commercial corn isolates, we carried out two-way analysis of variance (ANOVA), which allows for the use of both locus and population set as independent effects. Use of the same genetic markers across population sets results in increased statistical resolution for differences among populations because locus-to-locus variation is explicitly included in the model as opposed to in the error term as in a one-way analysis. In our case, we are most concerned with whether our population sets differed in allelic or genetic diversity; however, in all cases, the locus effect was significant, since loci differ in their levels of diversity regardless of population set. Therefore, we only report the *F*-values for the population effects. In terms of allelic diversity, FGIS samples (high, medium, low, and zero) harbored lower values for the number of different alleles (Na) when compared to commercial corn isolates (two-way ANOVA with least square (LS) mean differences Tukey test, population effect *F*_4,56_ = 13.50, *p* < 0.0001) (Figure 6). However, when evaluating measures that consider the distribution of alleles and diversity across the samples, the FGIS isolates demonstrated higher levels of genetic diversity across all remaining measures with the high-AF producers showing the highest levels of diversity overall. The FGIS collections had higher allelic diversity as reported by the number of effective alleles, Ne (*F*_4,56_ = 22.10, *p* < 0.0001), which represents an estimate of the number of equally frequent alleles and enables comparisons of allelic diversity across loci with diverse allele frequency distributions [31]. FGIS sets also had higher genetic diversity as measured by the information index, I (*F*_4,56_ = 23.05, *p* < 0.0001), which is equivalent to the Shannon–Weaver index of ecology. Lastly, the FGIS samples showed higher levels of unbiased haploid genetic diversity, u_h_ (*F*_4,56_ = 45.14, *p* < 0.0001). The lower genetic diversity in the commercial corn samples may be due to application of biocontrol products by some growers and prevalence of those particular genotypes within the associated *A. flavus* population.

Biological control with non-aflatoxigenic *A. flavus* is unique for several reasons. The first is the aspect that the intent of *A. flavus*-based biocontrol is not to prevent infection, but instead to simply to protect the commodity from becoming contaminated with the secondary metabolites of the toxigenic *A. flavus*. It may also be surprising that several successful, widely applied non-aflatoxigenic *A. flavus* products have been developed, while significant questions remain regarding the mechanism(s) of the biological control. Competitive exclusion of toxigenic isolates is commonly discussed and is certainly a component of the successful biocontrol [31], but persuasive evidence has also been presented for touch-based inhibition of AF, as well as AF suppression, through production of volatile signal molecules by the biocontrol isolates [32,33]. Furthermore, while there is research on the mechanism(s) of this biocontrol, there is also inquiry regarding the specificity of biocontrol, i.e., to what extent do biocontrol isolates need to align to the traits (e.g., mating type) of the toxigenic *A. flavus* population [27,34,35]. If many of the isolates from these high-AF corn samples were proven to be genetically distinct from the more common commercial corn samples, then it could be argued that this “super-virulent” or “super-toxigenic” genotype should be the target when selecting biocontrol agents or in testing improved, resistant corn hybrids. Instead, we noted great genetic diversity within these isolates and a similar distribution and level of genotypic clustering from a random sample of commercial corn, with a safer, more typical range of AF values. Thus, a prudent biocontrol screening program or corn breeding program would be advised to test against many genetically distinct toxigenic isolates.

## 3. Conclusions

Our objective in isolating *A. flavus* from naturally occurring, highly AF-contaminated (>50 ppb) corn was to better understand the characteristics of the *A. flavus* population that is actually causing economic loss and/or threatening the safety of food and feed. Analyses revealed this population to be more evenly divided between mating types 1-1 and 1-2 and incrementally more toxigenic than reported in another study of corn-associated *A. flavus* [16] and less sensitive to the fungicide azoxystrobin than another report [19]; however, overall, the population was not genetically distinct from another collection of *A. flavus* from commercial corn with more typical levels of AF contamination (average <50 ppb). The collection of isolates from high AF corn and from typical commercial corn both contained numerous isolates that were similar to, but slightly distinct from the commercial biocontrol isolate NRRL 21882, including AF-producing and AF-nonproducing isolates. The most unique aspect of the collection of isolates from high AF corn was the almost complete absence of exact NRRL 21882-matching genotypes, which were abundant in the other commercial corn samples and which may point to the efficacy of aflatoxin biocontrol using this isolate.

## 4. Materials and Methods 

### 4.1. Isolation from Contaminated Corn Samples

FGIS administers a monitoring program for evaluating the accuracy of official aflatoxin inspections. In this program, official service providers send samples representing official aflatoxin inspections to the Technology and Science Division (Kansas City, MO, USA) for reanalysis using the FGIS reference method. Less than 3% of these corn samples were contaminated with >10 ppb AF, and less than 1% had >20 ppb. Twenty-nine samples were encountered between 2019 and 2021 that had >50 ppb aflatoxin and were subject to further analysis, included here. These samples did not include identifying information. Milled grain samples were diluted with 0.1% Triton and isolated on MDRB medium [36], collecting five isolates per grain sample. Sporulating agar plugs from β-cyclodextrin-amended potato dextrose agar were stored in sterile water in screw-top vials at room temperature and archived for subsequent study. 

### 4.2. Characterization of A. flavus Isolates

A visual phenotype of each isolate was recorded including the production of many, small sclerotia with a sparse sporulation (S-type) or abundant sporulation and few, large sclerotia (L-type) [17] upon growth on β-cyclodextrin-amended potato dextrose agar. The colonies were scored for the pigmentation of the agar and fluorescence, which has been associated with aflatoxigenicity [18].

The method of Probst and Cotty [24] was modified to evaluate the AF-producing capacity of each isolate on corn. Five grams of cracked corn was adjusted to 25% moisture, autoclaved, and then inoculated and incubated for 6 days at 28 °C in 50 mL conical tubes with lids that were replaced with Tyvek covers. Then, 10 mL of 0.1% Triton X-100 was added and agitated to dislodge spores; a 1 mL aliquot sample was taken for DNA extraction [37], and 20 mL methanol added to the remainder. After further agitation, incubation overnight, and an additional agitation, an aliquot was taken for AF measurement via HPLC-FLD [38]. AF concentrations were binned as zero, low (1–19 ng/mL), medium (20–300 ng/mL), or high (>300 ng/mL) producers, consistent with Sweany et al. [16].

For genotyping of *A. flavus* isolates, SSR markers were amplified with 15 primer pairs [39,40] and amplicons separated on an ABI 373 DNA Analyzer with the LIZ 500 size standard. Amplifications were also run alongside a sample of *A. flavus* strain 21882 as a quality control/sizing standard. The mating type locus was amplified using the primers of Ramirez-Prado [41]. Alleles of the SSRs and MAT locus were scored with Geneious Prime (2002.2 www.geneious.com, accessed on 27 October 2022). The method of Ali et al. [19] was used to determine the sensitivity of each *A. flavus* isolate to the fungicide azoxystrobin. Testing was performed on potato dextrose agar and compared with potato dextrose agar supplemented with salicylhydroxamic acid (100 µg/mL) and azoxystrobin (2.5 µg/mL), which were determined to be the appropriate discriminatory dose. Plates were incubated in the dark, and the experiment was repeated twice. The radial growth rate of each isolate was used to determine the level of resistance to azoxystrobin. The criteria of Ali et al. [19] were used to characterize each isolate as sensitive (76–100% inhibition), reduced sensitivity (51–75% inhibition), moderately resistant (15–50% inhibition), and resistant (0–14% inhibition). 

### 4.3. Population Analyses

The *A. flavus* population collected in this study is highly unique, being isolated from a very narrow set of commercial corn samples contaminated with >50 ppb AF. The genotypes of the *A. flavus* population associated with this unusually contaminated corn were compared to the genotypes of *A. flavus* from more typical commercial US corn samples. Another survey of *A. flavus* from commercial corn samples (Mehl, in preparation) conducted from 2014 to 2017 collected over 3000 isolates and conducted SSR characterization as described above. From this collection, 150 isolates were randomly selected through use of a random number function for comparison to the population from highly AF-contaminated corn. Two of these random subsamples were analyzed to validate that the results described here are representative.

Measures of haplotypic diversity were calculated for each SSR locus using GenAlEx 6.5 [42] in Microsoft Excel including Na = number of alleles (Na), number of effective alleles (Ne = 1/(Sum p_i_^2^)), Shannon’s information index (I = −1 × Sum (p_i_ × Ln (p_i_)), diversity (h = 1 − Sum p_i_^2^), and unbiased diversity (u_h_ = (N/(N − 1)) × h), where p_i_ is the frequency of the i-th allele for the population, and Sum p_i_^2^ is the sum of the squared population allele frequencies. A principal coordinate analysis (PCoA) was also conducted on pairwise genetic distances using the standardized genetic distance method implemented in the program GenAlEx (Acton, Australia).

Differences among population sets in genetic diversity were assessed using two-way ANOVAs with the main effects of population sample (FGIS isolates were binned by AF production) and locus, with the dependent variables number of different alleles (Na), number of effective alleles (Ne), information index (I), and haploid genetic diversity (h). Use of the same genetic markers across population sets can result in increased statistical resolution for differences among populations because locus-to-locus variation is explicitly included in the model. Two-factor ANOVAs were followed with post hoc LS mean differences Tukey tests using JMP version 13 (SAS Institute, Cary, NC, USA).

## Figures and Tables

**Figure 1 toxins-14-00755-f001:**
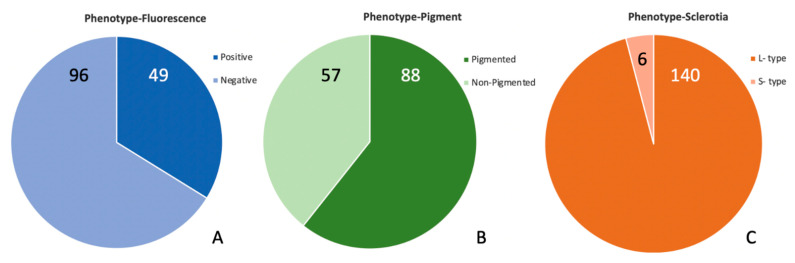
Visual phenotypes of *A. flavus* isolates from US corn samples with high aflatoxin contamination. (**A**,**B**) present the pigmentation and fluorescence of the growth medium (potato dextrose agar amended with β-cyclodextrin). The colony morphology in (**C**) indicates an S-type phenotype with numerous small sclerotia and limited sporulation or an L-type phenotype with abundant sporulation and either a few large sclerotia or no sclerotia present.

**Figure 2 toxins-14-00755-f002:**
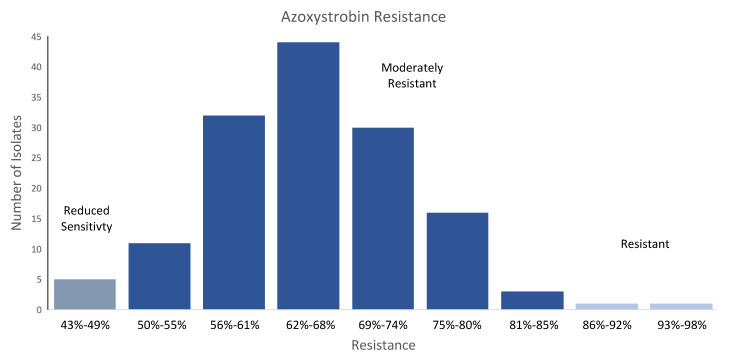
Resistance to azoxystrobin fungicide in isolates of *A. flavus* isolates from US corn samples with high aflatoxin contamination. All isolates were grown on potato dextrose agar amended with a discriminatory dose of azoxystrobin (2 µg/mL) and salicylhydroxamic acid (100 µg/mL). The criteria of Ali et al. (2020) were used to characterize each isolate as sensitive (0–24% resistant), reduced sensitivity (25–49% resistant), moderately resistant (50–85% resistant), or resistant (86–100% resistant).

**Figure 3 toxins-14-00755-f003:**
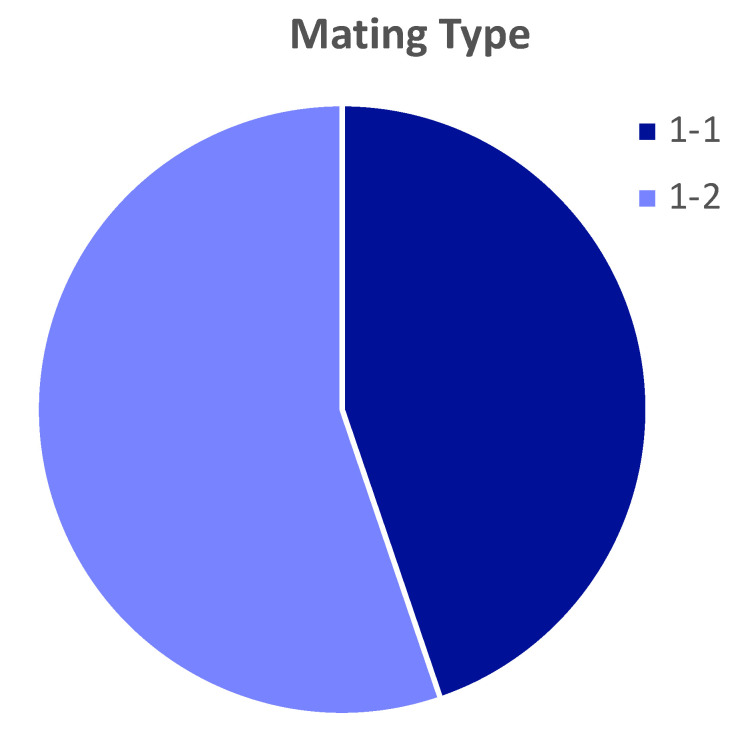
Mating type distribution of *A. flavus* isolates from US corn samples with high aflatoxin contamination. Mating types were determined by multiplex amplification of the MAT locus.

**Figure 4 toxins-14-00755-f004:**
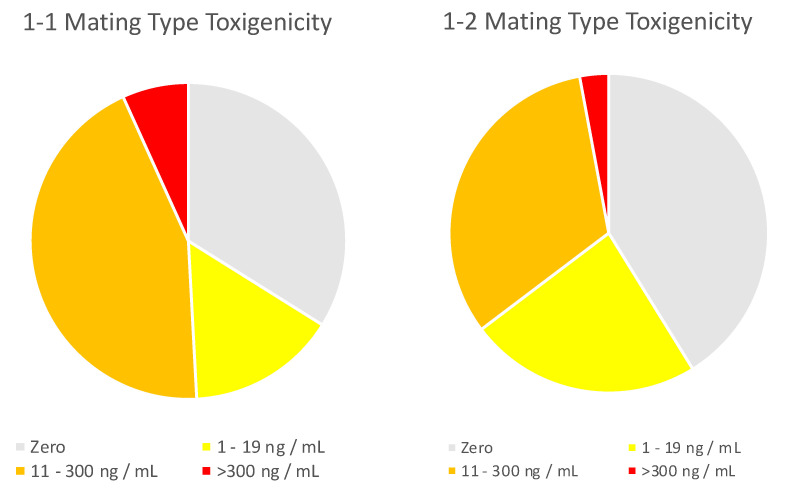
Distribution of non-aflatoxigenic (zero), low (1–19 ng/mL)-, medium (20–300 ng/mL)-, and high AF (>300 ng/mL)-producing isolates, separated by MAT genotype.

**Figure 5 toxins-14-00755-f005:**
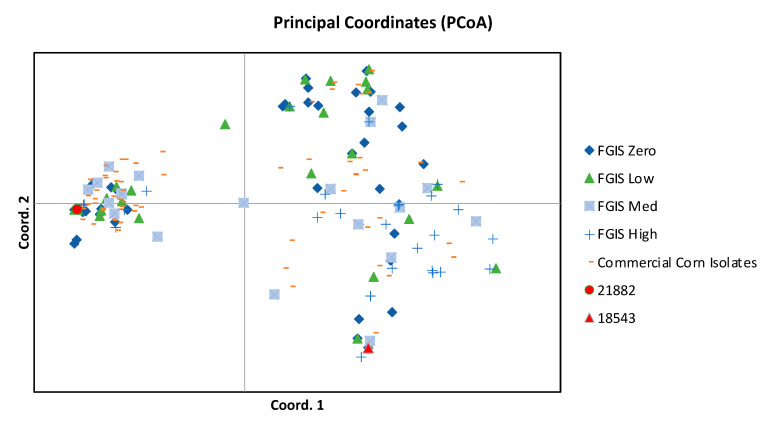
Principal coordinate analysis of *A. flavus* SSRs. Points designated as commercial corn isolates are based on the genotypes of isolates from a random subsample of *A. flavus* isolates from US commercial corn samples. Here, 21882 and 18543 are reference points based on the genotypes of two commercial biocontrol isolates. Principle coordinates 1 and 2 explain 31% and 9% of the variance, respectively. FGIS Zero, Low, Med, and High indicate isolates that are non-aflatoxigenic or that produced 1–19, 20–300, or >300 ng/mL aflatoxin, respectively.

**Figure 6 toxins-14-00755-f006:**
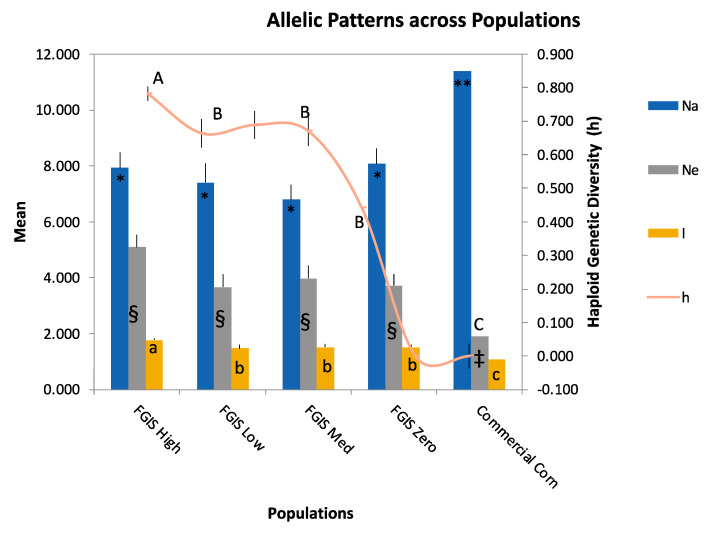
Comparisons of diversity between *A. flavus* isolates from highly aflatoxin contaminated (>50 ppb) corn (FGIS samples) and from commercial corn samples with typical levels of aflatoxin contamination (average <50 ppb). Na is the number of alleles. Ne is the number of effective alleles. Haploid genetic diversity is given as *h*. Shannon’s information index is given as *I*. Tukey’s mean separation test results given for each series with a capital letter (h), a lowercase letter (I), asterisks (Na), and § or ‡ (Ne). FGIS Zero, Low, Med, and High indicate isolates that are non-aflatoxigenic or produced 1–19, 20–300, or >300 ng/mL aflatoxin, respectively.

## Data Availability

The data presented in this study are available in Appendix A.

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
