# Peer review of "Characterization of the Aspergillus flavus Population from Highly Aflatoxin-Contaminated Corn in the United States"

_toxins, 2022, doi:10.3390/toxins14110755_

Round 1

Reviewer 1 Report

The article collected 29 highly contaminated corn samples and obtained 146 Aspergillus flavus isolates from these samples to characterize the visual phenotypes, aflatoxin producing potential and genotypes to explore the etiological cause of high aflatoxin contamination of US corn. The research was novelty, the methods were adequately described, and the results were clearly presente, while the style needs minor checked, i.e.: abbreviation of strain name in text and figure caption, space between numbers and units and so on.

Comment:

1. Please add the numbers of visual phenotypes of A. flavus in pie chart of Figure 1; Add the figure caption of FGIS Zero, FGIS LOW …….in Figure 5 and 6.

2. Please describe more sample information in 4.1. Isolation from contaminated corn samples, such as date.

3. Adequate discussion of the azoxystrobin resistance and SSRs were needed, i.e. 50% resistant was higher or lower than early data?

Author Response

Author response to Reviewer 1

The article collected 29 highly contaminated corn samples and obtained 146 Aspergillus flavus isolates from these samples to characterize the visual phenotypes, aflatoxin producing potential and genotypes to explore the etiological cause of high aflatoxin contamination of US corn. The research was novelty, the methods were adequately described, and the results were clearly presente, while the style needs minor checked, i.e.: abbreviation of strain name in text and figure caption, space between numbers and units and so on.

Comment:

  1. Please add the numbers of visual phenotypes of A. flavus in pie chart of Figure 1; Add the figure caption of FGIS Zero, FGIS LOW …….in Figure 5 and 6.

            Numbers added to charts in Fig. 1.  Text added to captions for Fig 5 and 6.

  1. Please describe more sample information in 4.1. Isolation from contaminated corn samples, such as date.

Text added at line 275-276 to include dates.  Also note re-wording at line 7.

  1. Adequate discussion of the azoxystrobin resistance and SSRs were needed, i.e. 50% resistant was higher or lower than early data?

            Resistance in the FGIS samples was greater than in the earlier data (Ali et al., Georgia peanut).  At line 132-134 we note: “50% of the Georgia isolates were “sensitive” or “reduced sensitivity” while none of the FGIS isolates were “sensitive”, 3% were “reduced sensitivity…”

Reviewer 2 Report

The MS needs to be substantially revised to make the manuscript more readable and consistent

L34 flatoxins-->A.flatoxins Please check the MS

L14 Abbreviations should be defined one time (first time) in the abstract and then one time from introduction and then use that abbreviation throughout the manuscript. I found too many mistakes regarding this issue.

192-197 Most of this is not highly relevant to you result as you didn’t look these common 21882-like isolates are derived from NRRL 21882

L320 Start a new paragraph. You may like to head it ' Statistical analysis '

L337+Reference

Use “Animals Journal” format throughout the manuscript, A minor revision is needed as per suggestion of referees.

Author Response

Author response to Reviewer 2

The MS needs to be substantially revised to make the manuscript more readable and consistent

            Note the restatement of the objective and hypotheses at the end of the revised introduction.  This should make the work easier to follow.  Many of the inconsistencies have been addressed by reformatting the references. 

L34 flatoxins-->‘A.flatoxins’ Please check the MS.

            The reviewer may have mistaken the common name of this mycotoxin for a mis-formatted Latin binomial.  The text at line 34 is correct.

L14 Abbreviations should be defined one time (first time) in the abstract and then one time from introduction and then use that abbreviation throughout the manuscript. I found too many mistakes regarding this issue.

            The manuscript has been reviewed for this and we have added definitions in the abstract.

192-197 Most of this is not highly relevant to you result as you didn’t look these common 21882-like isolates are derived from NRRL 21882 

            This is just an observation for the discussion.  We just note a likely explanation for a pattern seen in our data and the data of others.

L320 Start a new paragraph. You may like to head it ' Statistical analysis '

            Section 4.3 has been renamed ‘Population analyses’.  A new paragraph has been started at what is now Line 324

L337+Reference

Use “Animals Journal” format throughout the manuscript, A minor revision is needed as per suggestion of referees.

            The author’s guide for toxins indicates flexibility on the format, as long as the document is consistent.

Reviewer 3 Report

The general impression on this manuscript is that it describes a remarkable analytical work which, however, does not rest on solid assumptions.

The Authors seem to draw conclusions based on the results of analyses without having clearly expressed the hypotheses to be demonstrated with the implementation of the laboratory or statistical technologies employed. For each hypothesis declared, the underlying rationale for verifying it should then be highlighted, that is the logical reasoning or the specific evidence from the literature that explains how the application of the aforementioned technologies can verify the declared hypotheses.

A comparative control, say "C", group (where the aflatoxin concentration was less than 50 ppb) was not set up to compare the outcomes, related to aflatoxin production, between the high concentration, say"H", aflatoxin group and C. If the outcomes were similar between C and H, it would not have been possible to speak of characterization unless the features used for the characterization had already been demonstrated effective in the literature, and this, in my opinion, is not clear from the manuscript.

Moreover the following particular points.

Abstract: the expression "similar genotypic diversity" is unclear.

Line 185: general genetic similarity is not necessarily a factor characterizing the presence of a similar degree of aflatoxicity. Genetic similarities are needed in relation to specific loci.

Lines 282-284: the positive relationship between a type of sclerotia production and aflatoxigenicity does not seem fully justified as the substrate analyzed in the present work is very different from the article cited.

Lines 265-268: In my opinion, this conclusion indirectly demonstrates, more than anything else, the "pesticidal" efficacy of A. flavus NRRL 21882, which is present in most of the samples, that are those with little aflatoxigenicity.

In my opinion the Authors should revise the manuscript taking care of the points previously mentioned.

Author Response

Author response to Reviewer 3

The general impression on this manuscript is that it describes a remarkable analytical work which, however, does not rest on solid assumptions.

The Authors seem to draw conclusions based on the results of analyses without having clearly expressed the hypotheses to be demonstrated with the implementation of the laboratory or statistical technologies employed. For each hypothesis declared, the underlying rationale for verifying it should then be highlighted, that is the logical reasoning or the specific evidence from the literature that explains how the application of the aforementioned technologies can verify the declared hypotheses.

A comparative control, say "C", group (where the aflatoxin concentration was less than 50 ppb) was not set up to compare the outcomes, related to aflatoxin production, between the high concentration, say"H", aflatoxin group and C. If the outcomes were similar between C and H, it would not have been possible to speak of characterization unless the features used for the characterization had already been demonstrated effective in the literature, and this, in my opinion, is not clear from the manuscript.  [

            In fact, a comparative group has been included in the genetic comparisons.  The genotypic comparisons are between the “H” population (FGIS isolates) and a “C” population, randomly selected from commercial corn samples.  The objective statement in the introduction has been rewritten and should make this clear.

Moreover the following particular points.

Abstract: the expression "similar genotypic diversity" is unclear.

            That wording is no longer included.

Line 185: general genetic similarity is not necessarily a factor characterizing the presence of a similar degree of aflatoxicity. Genetic similarities are needed in relation to specific loci. 

            Text added at line 185 noting that some of the similarity to NRRL 21882 was found in aflatoxigenic and non-aflatoxigenic isolates. 

Lines 282-284: the positive relationship between a type of sclerotia production and aflatoxigenicity does not seem fully justified as the substrate analyzed in the present work is very different from the article cited.

            The reference cited here (#18, Abbas et al., 2004) used ß-cyclodextrin amended potato dextrose agar, just as in the present study.  The reference to sclerotia type (#17 Cotty, 1994) at line 290, used a different method (5/2 V8 agar), but identification of S-type morphology is possible on either medium. 

Lines 265-268: In my opinion, this conclusion indirectly demonstrates, more than anything else, the "pesticidal" efficacy of A. flavus NRRL 21882, which is present in most of the samples, that are those with little aflatoxigenicity.

            We agree.  Text added to the conclusion that echoes this point.

In my opinion the Authors should revise the manuscript taking care of the points previously mentioned.
